# Inpatient Mental Healthcare before and during the COVID-19 Pandemic

**DOI:** 10.3390/healthcare9121613

**Published:** 2021-11-23

**Authors:** Alan B. McGuire, Mindy E. Flanagan, Marina Kukla, Angela L. Rollins, Laura J. Myers, Emily Bass, Jennifer M. Garabrant, Michelle P. Salyers

**Affiliations:** 1Health Services Research & Development, Richard L. Roudebush VAMC, Indianapolis, IN 46202, USA; meflanag@iupui.edu (M.E.F.); marina.kukla@va.gov (M.K.); angela.rollins@va.gov (A.L.R.); laura.myers2@va.gov (L.J.M.); emibass@iu.edu (E.B.); jwilkers@iu.edu (J.M.G.); 2Department of Psychology, Indiana University-Purdue University Indianapolis, Indianapolis, IN 46202, USA; mpsalyer@iupui.edu; 3Health Services Research, Regenstrief Institute, Indianapolis, IN 46202, USA; 4Division of General Internal Medicine and Geriatrics, Indiana University School of Medicine, Indianapolis, IN 46202, USA

**Keywords:** mental health, COVID-19, hospital care, service disruption

## Abstract

Prior studies have demonstrated disruption to outpatient mental health services after the onset of the COVID-19 pandemic. Inpatient mental health services have received less attention. The current study utilized an existing cohort of 33 Veterans Health Affairs (VHA) acute inpatient mental health units to examine disruptions to inpatient services. It further explored the association between patient demographic, clinical, and services variables on relapse rates. Inpatient admissions and therapeutic services (group and individual therapy and peer support) were lower amongst the COVID-19 sample than prior to the onset of COVID-19 while lengths of stay were longer. Relapse rates did not differ between cohorts. Patients with prior emergent services use as well as substance abuse or personality disorder diagnoses were at higher risk for relapse. Receiving group therapy while admitted was associated with lower risk of relapse. Inpatient mental health services saw substantial disruptions across the cohort. Inpatient mental health services, including group therapy, may be an important tool to prevent subsequent relapse.

## 1. Introduction

COVID-19 wreaked havoc on the healthcare system. Some emergency departments and intensive care units were flooded with patients [1] and early indicators predicted demand beyond hospital capacity [2] while other non-COVID-19 admissions fell dramatically [3]. In addition, adult mental health services saw significant changes. Holland and colleagues [4]) report ED visits related to mental illness, suicide, substance abuse and other mental health crises increased during the primary US surges from March to October 2020 compared to the same period in 2019. In the UK, Chen and colleagues [5] reported an overall decrease in demand and provision of outpatient mental health services early in the pandemic, followed by a subsequent increase. Titov and colleagues [6] reported on an initial surge in virtual mental healthcare demand. Finally, Ng and colleagues [7] reported antidotally acute exacerbations in patients with obsessive-compulsive disorder and personality disorders. While these studies provide some understanding of changes to outpatient mental health services, less is known about inpatient mental health services.

In one study of seven Italian mental health units, inpatient mental health admissions dropped markedly in the 40 days immediately following the onset of COVID-19 in the region [8]. Factors specific to inpatient mental healthcare make managing both the pandemic and mental health services challenging [9]. Shifts in inpatient bed allocations and enhanced precautions to prevent disease spread are likely to affect access to inpatient mental healthcare in general, as well as access to inpatient care elements such as individual and group therapy for those who are hospitalized. Services provided during inpatient stays are designed to stabilization symptoms, prepare patients to cope with their mental illness post-discharge, and link patients with outpatient mental health treatment. Taken together, it is vital to understand the impact of COVID-19-related acute inpatient mental health services disruption on patients.

Relapse is a common aspect of mental illness and can result in repeated inpatient stays [10] that place a substantial burden on the healthcare system and negatively impact patients’ quality of life. Psychiatric relapses also result in frequent emergency department (ED) visits, with some studies pointing toward a disproportionate share of ED visits and costs from high utilizers, including those with mental health and substance use disorders [11,12]. Some research has examined drivers of relapse, including patient factors (e.g., illness severity) and the role of outpatient services following inpatient care [13,14,15,16,17]. However, little is known about systemic factors in relapse. Examining the impact of the COVID-19 within a system of care can provide important insights into how disruptions in inpatient services may affect relapse.

To understand the impact of the pandemic on inpatient mental healthcare, the current study included the following aims: Aim 1—describe changes to inpatient mental health services, including admissions, length of stay, and therapeutic services, before and after the COVID-19 pandemic onset; and Aim 2—examine the relationship between inpatient mental health service provision and relapse rates. In an exploratory fashion, we also investigated the relationship between patient-level factors and relapse. To accomplish these objectives, we built on a pre-existing cohort of 34 Veterans Health Affairs (VHA) acute inpatient mental health units [18]. Analyses compare inpatient services before and after the issuance of the Veterans Health Affairs COVID-19 Response Plan [19]. Although the policy stated, “Mental health, medical, and surgical specialty consultation should be conducted using non-face-to-face methods (e.g., telehealth, telephone, and e-consults), when possible,” it did not clarify functionality specific to acute, inpatient mental health services. Nonetheless, its issuance divides services into two reasonable cohorts of inpatient admissions—prior to COVID-19 and during COVID-19.

## 2. Materials and Methods

### 2.1. Sample

The initial study, Recovery-oriented Acute INpatient Mental Healthcare (RAIN-MH), focused on the implementation of recovery-oriented acute inpatient mental health services. The cohort included a nationally representative set of 34 acute inpatient units from every region of the continental United States, including urban and rural sites, with over 10,000 patients treated on these units [18]. Data for the current study were extracted for patients with an inpatient psychiatric admission from 1 September 2019 to 15 March 2021 on participating units; one closed unit was excluded from analyses, leaving a site sample of 33.

### 2.2. Measures

Data from the Veterans Affairs (VA) Corporate Data Warehouse (CDW) (The Department of Veterans Affairs (VA), Office of Information & Technology) were used to identify patients hospitalized on an acute mental health ward at one of the 33 VA facilities included in the original study. The CDW data include information from the VA electronic medical record that is used across the nationwide VA system and includes both clinical and administrative functionality [20]. The CDW data included inpatient and outpatient data files (e.g., clinical encounters with associated diagnostic and procedural codes) during the acute inpatient stay and in the time before and after the inpatient stay. For Aim 1, we abstracted information for each admission to characterize key inpatient services, including length of stay, total number of group and individual therapy sessions, and peer support service encounters. Group and individual therapy were identified using Current Procedural Terminology (CPT) codes. (Appendix A). Relapse (Aim 2 outcome) was defined as a mental health or substance abuse-related ED visit or mental health-related admission in the 30 days following discharge. Demographics (age, sex, race, and ethnicity), substance use disorder diagnosis, personality disorder diagnosis, rurality, suicide flag present in the medical record, and distance from patient home address to the Veterans Affairs Medical Center (VAMC) were also obtained as potential covariates.

For Aim 1, to describe changes to inpatient mental health services before and after the COVID-19 pandemic onset, we collated inpatient services data over time relative to the number of patients admitted. Specifically, inpatient stay and patient data elements were aggregated for each site by month to calculate total number of admissions and rates of individual therapy and group therapy sessions. Group therapy rates were calculated as total number of group therapy encounters for each patient during the inpatient admission divided by total number of patient-hospital days per month. The same calculation was made for individual therapy rates

### 2.3. Analyses

Descriptive statistics were calculated for all measures. Plots were created for number of admissions, rates of group and individual therapy, percent change in number of admissions and group therapy rates by month and by site. Because disruption in admissions and group therapy was most pronounced, we calculated the percent change for these variables for each month April 2020 to March 2021 and compared this to a baseline period (1 April 2019–29 March 2020) to compare changes during the pre-COVID-19 and COVID-19 periods. For patients with multiple admissions, one admission was randomly selected for Aim 1 analysis (using simple random selection). To account for the possibility that a randomly selected sample of admissions was not representative of the entire sample, this selection process was used to create 50 samples. Analyses were conducted on all 50 samples and calculated statistics averaged. To compare patient characteristics for inpatients prior to VHA COVID-19 policy release on 23 March 2020, and after, bivariate comparisons were conducted using independent *t*-tests for continuous variables and chi-square tests of independence for categorical variables. Additionally, a logistic model was estimated to test the impact of timing of admission on relapse (admission prior to 23 March 2020 or 23 March 2020 and later). A logistic regression model was estimated to test the impact of patient-level predictors and service disruption on relapse (for inpatient stays April 2020 to March 2021). A mixed-effects model was evaluated, and the site-specific effects were nearly 0. The intraclass correlation was approximately 1% so the mixed-effects model was deemed unnecessary.

Procedures were approved by the Institutional Review Board of Indiana University Purdue University Indianapolis (protocol code 1704043216, approved 6 July 2017).

## 3. Results

### 3.1. Pre-COVID-19 Period and COVID-19 Period: Veteran Characteristics and Inpatient Services

The full sample contained 17,374 patients and 25,757 admissions. The majority of patients had one admission (75.6%), which accounted for 51.0% of admissions. As shown in Table 1, patient characteristics were largely similar from pre-COVID-19 and COVID-19 samples. However, compared to pre-COVID-19, the COVID-19 sample contained more patients without prior mental health-related admissions, and patients resided further from the VA hospital.

The COVID-19 sample, as compared to the pre-COVID-19 sample, received a lower rate of inpatient services (group therapy, individual psychotherapy, and peer support), had longer inpatient stays, and received fewer outpatient mental health visits within 7 days post-discharge (Table 2). Because gross change is most striking, the overall month-to-month percent change in admissions and group therapy rates are shown in Figure 1 and Figure 2 for illustrative purposes. Admissions show a noticeable drop in April 2020 and again in November and December 2020, which correspond to peaks in COVID-19 infections rates in the United States. Relapse rates were just above 10% for both the pre-COVID-19 and COVID-19 periods.

### 3.2. Predictors of Relapse

Timing of admission (pre-COVID-19 or during COVID-19) was not associated with relapse (Table 3). Given the differences in patient mix before and after COVID-19 onset, we calculated risk-adjusted odds of relapse excluding inpatient services variables. The adjusted odds ratio for relapse during COVID-19 was 1.10 [95% CI = 0.99, 1.22] in comparison to the pre-COVID-19 sample.

While veteran demographic characteristics were not associated with relapse, prior emergent service usage (previous ED visits and admissions) and diagnostic factors (presence of a personality or substance abuse disorder) were associated with increased risk for relapse. Even when controlling for these variables, receipt of group therapy while hospitalized was still associated with a reduced risk of relapse.

## 4. Discussion

The current study examined service disruption in acute inpatient mental healthcare at 33 VHA facilities across the continental United States. This study joins a growing number of analyses demonstrating disruption to mental health services during the COVID-19 pandemic. While previous studies have demonstrated fluctuations in utilization of outpatient and emergency mental healthcare during the pandemic [13,20,21] and inpatient admissions [15], to our knowledge, this is the first study to demonstrate disruption to inpatient services. More specifically, we found patients were less likely to receive pillar services of inpatient mental health—individual therapy, group therapy, and peer support services following the onset of the pandemic. Additionally, patients stayed on inpatient units longer and were less likely to receive timely outpatient care following discharge. Notably, disruptions and resulting periods of rebounding varied widely across sites, making it difficult to ascertain a clear-cut period of “recovery” from COVID-19 disruptions in this national data set. Given the rise of the delta variant and infection and hospitalization rates in summer of 2021, the notion of a “stable” period of national recovery from COVID-19 disruptions may be premature.

Although the current analysis cannot provide reasons for the disruption to therapeutic inpatient services, several factors likely contribute. Staffing shortages have clearly affected a full range of healthcare services [22,23] leading to fewer available to provide services, such as individual and group therapy. In addition, risk mitigation strategies may have affected care. For instance, the necessity of social distancing may have resulted in fewer group services. Additionally, outpatient providers (e.g., group leaders and peer support specialists) who normally provide some services on inpatient units might have been prohibited from entering the unit or may have been distanced from the unit (e.g., triaged to telehealth only). Best practices, resources, and logistics of providing telehealth group and individual therapy on inpatient units is in its infancy and worthy of further study.

This summary across all sites does not reveal the widely varied experience of the many sites under consideration. Despite being in one system of care, the variability across sites was high, ranging from one site with little disruption in admissions (dropping by only 3%) to one site with a 100% reduction (apparent shutdown) in April 2020 This wide variability was apparent in access to group therapy sessions. In some sites, access to group therapy was virtually unchanged during the COVID-19 pandemic (Figure A1). In other sites, there were temporary month-to-month declines that were not apparently linked to VHA COVID-19 policy (Figure A2), while in some it was noticeably reduced and did not recover (Figure A3).

Patient characteristics were largely unchanged during the pandemic, with two exceptions. First, the distance between patients’ residence and the hospital increased. This change could represent the need to travel farther for inpatient services if other inpatient units were at capacity due to COVID-19 cases or diversion of staffing. Distance to a VAMC might also serve as a marker for risks of social isolation, a clinical phenomenon associated with known negative impacts on a variety of well-being indicators [24,25]. The second key difference in patient populations during the pandemic is that patients were less likely to have been hospitalized previously. It is unclear what drove this trend; however, it is possible that the multiple stressors of the pandemic (e.g., fear of infection, stress of illness of self and close others, and economic turmoil) may have driven mental health crisis in patients who had heretofore coped with underlying mental health concerns without requiring acute inpatient services.

While relapse rates did not differ significantly between the pre-COVID-19 and during COVID-19 samples, several potential explanations exist which differ in their implications for mental healthcare. Most optimistically, it may be that the decrease in inpatient mental health services, in general (decreased admissions) and in therapeutic services provided during admissions (group, peer services), did not impact patients’ tendency toward relapse, or that any impact was moderated by other factors (e.g., increased access to online mental health services) [14]. Alternately, there are reasons to be concerned. For instance, Leff and colleagues (2021) found mental health presentations to a pediatric emergency department to drop over 60% immediately following the onset of the pandemic (March to May 2020), indicating an overall reticence for persons in crisis to seek emergency care, perhaps due to fears of COVID-19 exposure or impressions of emergency services being overburdened by the pandemic. Given these findings, it is possible that patients in our study may have avoided a return visit for crisis services, even when potentially needed.

The current study demonstrates ongoing patient-level risks for relapse, irrespective of COVID-related factors. In particular, our study joins prior literature in showing patients with a history of previous crisis services as well as persons with substance use disorders to be at higher risk for relapse [26,27,28]. Importantly, to our knowledge, this is the first study to demonstrate patients receiving group therapy services to be at a lower risk for relapse. Several potential explanations are possible. First, amongst the purposes of group services is to provide skills and resources to prevent future relapse. However, group services on these units are voluntary and the relationship may be spurious—patients who are generally better functioning may be more likely to both attend group and avoid relapse. This explanation is somewhat undercut as our analyses found the relationship to remain even when controlling for previous ED visit, inpatient admissions, and diagnoses. Future research is needed to explore the protective effects of groups services on relapse and other important outcomes.

These findings should be viewed in light of limitations to this study. This study did not include a priori hypotheses due to the early and emerging nature of our understanding of the disruptions to healthcare caused by the COVID-19 pandemic. Future research is needed to test hypotheses. Moreover, the current data are derived from a subset of inpatient units in one, large and partially integrated healthcare system. Therefore, results may not generalize to other settings. Relatedly, patients may relapse and use services outside of the VHA system, which would not be recorded in the current data. Additionally, while numerous inpatient treatments were measured and accounted for, other services, including medications, were not included in the current analyses. Future research would benefit from a more comprehensive set of inpatient treatment variables. Finally, there are numerous ways to operationalize the onset and burden of the COVID-19 pandemic. For instance, one could assess overall case rates at the regional or national level, case rates within the hospital system, or percentage of inpatient beds filled with COVID-19 patients. The current study utilized one date of onset for the entire sample—fixed to administrative action intended to reduce infection risk—but alternate operationalizations should be explored in future research.

## 5. Conclusions

Inpatient mental healthcare services were substantially disrupted during the COVID-19 pandemic. Despite these disruptions, relapse rates did not change from prior to the pandemic’s onset. Even accounting for patient-level factors such as diagnosis and prior hospitalizations, patients who received group therapy during their inpatient stay were less likely to relapse.

## Figures and Tables

**Figure 1 healthcare-09-01613-f001:**
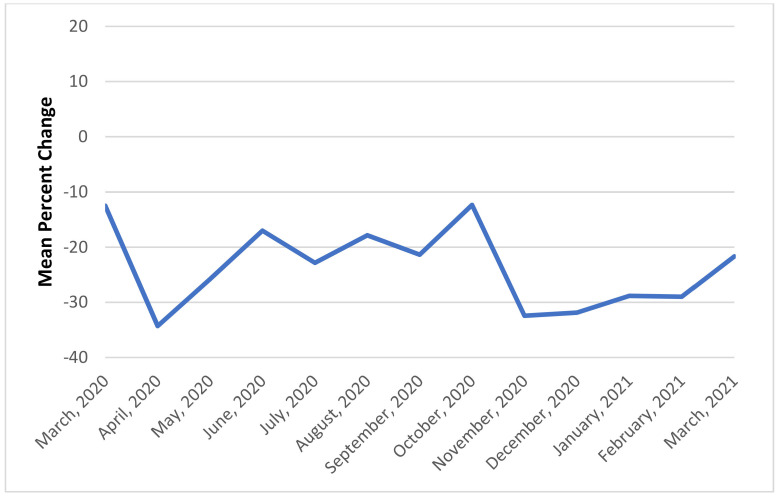
Mean percent change in admissions.

**Figure 2 healthcare-09-01613-f002:**
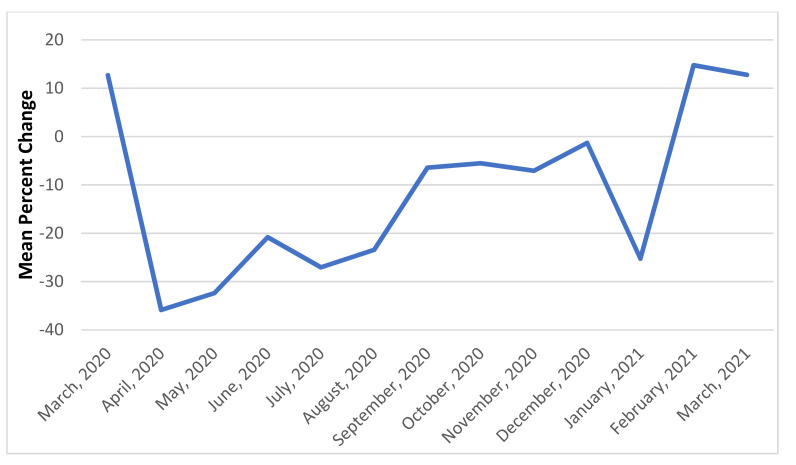
Mean change in group therapy rates per patient.

**Table 1 healthcare-09-01613-t001:** Descriptive statistics of veteran characteristics for 50 randomly drawn samples with one observation per veteran, before and after VHA COVID-19 policy mandate (*n* = 17,374).

Characteristic	Pre-COVID-19 Period Admission 1 September 2019–Discharge 21 February 2020		COVID-19 Period Admission 23 March 2020–Discharge 15 March 2021		*t*-Test or Chi-Square Statistic (Mean per 50 Replications)
	Mean or Count	SD or Percent	Mean or Count	SD or Percent	
Age (years), mean (SD)	49.92	14.67	49.60	14.96	t (15,545) = 1.37, *p* = 0.19
Gender (male), *n* (%)	6212.3	87.00	8885.70	86.83	Χ^2^(1) = 0.18, *p* = 0.72
Race, *n* (%): Black/Other/Unknown White	2339.62 4800.56	32.77 67.23	3335.38 6898.44	32.59 67.41	Χ^2^(1) = 0.15, *p* = 0.75
Ethnicity, *n* (%): Hispanic or Latino Not Hispanic or Latino	642.96 6247.40	9.33 90.67	949.32 8711.66	9.83 90.17	Χ^2^(1) = 1.23, *p* = 0.31
Distance from hospital *	69.54	209.51	77.99	232.76	t (16291) = −2.50, *p* = 0.02
Geographic Classification, *n* (%): Urban Rural Highly Rural	6394.84 728.74 16.60	89.56 10.21 0.23	9140.22 1074.60 19.00	89.31 10.50 0.19	Χ^2^(2) = 0.99, *p* = 0.63
Substance use (yes), *n* (%)	4889.60	68.48	7139.10	69.76	Χ^2^(1) = 3.30, *p* = 0.08
Personality disorder (yes), *n* (%)	1049.96	14.70	1431.84	13.99	Χ^2^(1) = 1.91, *p* = 0.22
Previous MH admission (yes) *, *n* (%)	1637.38	22.93	1915.34	18.72	Χ^2^(1) = 46.19, *p* < 0.001
Previous ED visit (yes), *n* (%)	1710.54	23.96	2333.32	22.80	Χ^2^(1) = 3.28, *p* = 0.10
Suicide Flag (yes), *n* (%)	2005.26	28.08	2916.30	28.47	Χ^2^(1) = 0.44, *p* = 0.59

* Significant at *p* < 0.05.

**Table 2 healthcare-09-01613-t002:** Descriptive statistics of service usage for 50 randomly drawn samples with one observation per veteran, before and after VHA COVID-19 policy mandate (*n* = 17,374).

Characteristic	Pre-COVID-19 Period Admission 1 September 2019–Discharge 21 February 2020		COVID-19 Period Admission 23 March 2020–Discharge 15 March 2021		*t*-Test or Chi-Square Statistic (Mean per 50 Replications)
	Mean or Count	SD or Percent	Mean or Count	SD or Percent	
Length of stay *	7.44	9.00	8.11	11.18	t (17017) = –4.36, *p* < 0.001
Individual Psychotherapy (yes), *n* (%)	4104.92	57.49	5639.92	55.11	Χ^2^(1) = 9.78, *p* = 0.003
Group Therapy (yes), *n* (%) *	5067.10	70.97	5814.18	56.81	Χ^2^(1) = 360.04, *p* < 0.001
Peer support (yes), *n* (%) *	1893.06	26.51	2064.84	20.18	Χ^2^(1) = 96.14, *p* < 0.001
MH clinic visits, 7 days post-discharge, mean (SD)	2.55	2.73	2.33	2.54	t (14625) = 5.54, *p* < 0.001
Relapse ^1^, *n* (%)	738.10	10.34	1090.16	10.65	Χ^2^(1) = 0.83, *p* = 0.50

^1^ Relapse operationalized as ED visit or mental health-related admission 30 days post-discharge. * Significant at *p* < 0.05.

**Table 3 healthcare-09-01613-t003:** Summary results from logistic regression predicting 30-day relapse after discharge (*n* = 17,374 with 50 replications).

Effect	Estimate	Standard Error	DF	Wald Chi-Square	*p*-Value
Intercept	–2.83	0.12	1	535.53	<0.001
Race (Not White)	–0.01	0.05	1	0.40	0.63
Age	0.003	0.002	1	3.57	0.11
Previous ED Visit (yes)	0.28	0.02	1	136.11	<0.001
Previous MH Admission (yes)	0.38	0.03	1	217.10	<0.001
Length of Stay	–0.0005	0.003	1	0.35	0.64
Personality Disorder (yes)	0.20	0.07	1	9.27	0.02
Substance Use (yes)	0.34	0.06	1	31.79	<0.001
Individual Psychotherapy (yes)	–0.03	0.05	1	0.96	0.48
Group Therapy (yes)	–0.17	0.05	1	9.92	0.009
Admission after 23 March 2020	0.06	0.05	1	1.91	0.30

## Data Availability

Restrictions apply to the availability of these data. Data were obtained from the VA Corporate Data Warehouse [CDW]. Data must remain on Department of Veterans Affairs servers. Investigators interested in using these data for analyses should email the corresponding author.

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
