# Peer review of "Inpatient Mental Healthcare before and during the COVID-19 Pandemic"

_healthcare, 2021, doi:10.3390/healthcare9121613_

Round 1

Reviewer 1 Report

I thank the authors for the article. There are certainly some interesting trends here, although the writing and discussion require more in-depth exploration.

Specific comments:

  1. Anecdotally, some authors have reported increased numbers of patients with obsessive-compulsive disorder (OCD) or personality difficulties seeking psychiatric help in the recent months. Predictably, their conditions may be exacerbated by the fear of contagion and of loved ones falling ill or feelings of emptiness when quarantined from others (citation: pubmed.ncbi.nlm.nih.gov/32380875). This should be mentioned in the introduction.
  2. Why the specific focus on group therapy rates? This remains unclear to the general reader.
  3. The percentage change calculated is rather problematic. It is highly prone to confounders and temporal bias. The authors need to better address this.
  4. Rather than present the absolute numbers of cases, it should be presented as an incidence rate per 1000 person-years for ease of comparison.
  5. The y-axes in Figure 1 (should be labelled as (a) and (b) respectively) are missing.
  6. Suggest to add a conclusion section to summarise the key findings.

Author Response

  1. Anecdotally, some authors have reported increased numbers of patients with obsessive-compulsive disorder (OCD) or personality difficulties seeking psychiatric help in the recent months. Predictably, their conditions may be exacerbated by the fear of contagion and of loved ones falling ill or feelings of emptiness when quarantined from others (citation: pubmed.ncbi.nlm.nih.gov/32380875). This should be mentioned in the introduction.

Thank you for point out this article. We have included this citation.

  1. Why the specific focus on group therapy rates? This remains unclear to the general reader.

Group therapy rates are the focus (beyond other variables that are significantly associated with relapse) in the Figures. We now clarify this is because the change in group therapy (and admission rates) is most striking in its gross percent change, and therefore are displayed for demonstration purposes.

  1. The percentage change calculated is rather problematic. It is highly prone to confounders and temporal bias. The authors need to better address this.

Thank you for this comment. We believe this comment pertains to our selection of the baseline period. Using the average rate across 3-months (December, 2019 – February, 2020) as a baseline comparison was our attempt at reducing any influence of temporal bias. In reflecting on the landmark study by Campbell (1968), temporal bias is critical to address to avoid distortion due to selection of timepoints for comparison. However, here, we ameliorated any distortion by averaging 3-months for the baseline comparison. Also, percentage change is not used as an outcome, but rather to detect patterns, which were not significant. Our primary findings are that the percentage change was idiosyncratic, not revealing a robust pattern across facilities.

  1. Rather than present the absolute numbers of cases, it should be presented as an incidence rate per 1000 person-years for ease of comparison.

We believe that the comparison of percetages between pre-COVID and post-COVID is most accessible to readers, so do not agree that re-scaling the rates to 1000 person years would result in greater understanding. The rates of relapse (10.34%, 10.65%) are not meaningfully different, which is the primary finding.

  1. The y-axes in Figure 1 (should be labelled as (a) and (b) respectively) are missing.

These figures are now Figures 1 and 2 and y-axes have been appropriately labeled.

  1. Suggest to add a conclusion section to summarise the key findings.

We thank the reviewer for this recommendation. We added such a section.

Reviewer 2 Report

It is a very good and interesting study on a subject that can hardly be adequately explored.

I would ask the authors to make the following changes:

- Institutional Review Board Statement:  from the end of the text to be moved to the method.

-Line 101 and elsewhere (For Aim 1….) please rewrite the aim in detail.

-Table 1 has very interesting data but it is a lot, it would be better to split it into two tables.

- Appendix A: Figure 1 and Figure 2 are important I would prefer to transfer them to the results.

- It is important to add Limitations, for example the parameter of drug treatment was not considered. 

Author Response

- Institutional Review Board Statement:  from the end of the text to be moved to the method.

We have added this language to the methods.

-Table 1 has very interesting data but it is a lot, it would be better to split it into two tables.

We have separated data into two tables- Table 1 includes Veteran characteristics and Table 2 service usage.

- Appendix A: Figure 1 and Figure 2 are important I would prefer to transfer them to the results.

We now include the figures as part of the results.

- It is important to add Limitations, for example the parameter of drug treatment was not considered. 

We now address limitations in our discussion, including the lack of inclusion of medication data in our dataset.

Reviewer 3 Report

This study highlights the impact of COVID-19 pandemic on inpatient mental health services by examining an existing cohort of 33 Veterans Health Affairs (VHA) acute inpatient mental health units. The manuscript focuses on the role of mental health services in preventing relapse of mental disorders. Although the manuscript is well organized with a very good methods, there are some issues that do not encourage a publication in the current form.

Introduction. The introduction could be better structured. A general overview on COVID-19 effects on healthcare should be put first, followed by an insight on mental healthcare, not the opposite.

A direct description of the main hypothesis of the study is lacking. At last, a more thorough description of the aims of the study should be included.

The reasons why the authors pointed out the issues underlined in lines 63-70 of page 2 are not clear.

Materials and methods. “Aim 1”, described in line 90 should be better clarified and specified, as it has not been done in the previous parts of the article, therefore such an abbreviation does not help the reader understand intuitively what “aim 1” means.

Discussion. Figure captions should be better stated.

The significant intercept effect from logistic regression predicting 30-day relapse after discharge could be highlighted and discussed.
The sentences in lines 213-217 could be moderated since they are too speculative, and unsupported by the literature in the manuscript.

Author Response

Introduction. The introduction could be better structured. A general overview on COVID-19 effects on healthcare should be put first, followed by an insight on mental healthcare, not the opposite.

We have made this change.

A direct description of the main hypothesis of the study is lacking. At last, a more thorough description of the aims of the study should be included.

A more thorough description of the study aims has been included in the introduction. Because research on the impact of the COVID-19 pandemic is still early and emerging, the current study was not hypothesis drive, but rather descriptive. We include this issue in the limitations section.

The reasons why the authors pointed out the issues underlined in lines 63-70 of page 2 are not clear.

These lines provide important information about how we have operationalized the pandemic onset in the current analysis. There are several ways to operationalize the pandemic (nationally and locally). We include a discussion of this issue in the limitations section.

Materials and methods. “Aim 1”, described in line 90 should be better clarified and specified, as it has not been done in the previous parts of the article, therefore such an abbreviation does not help the reader understand intuitively what “aim 1” means.

The aims have been revised for greater clarity and detail.

Discussion. Figure captions should be better stated.

We agree and now include captions that better describe the data presented.

The significant intercept effect from logistic regression predicting 30-day relapse after discharge could be highlighted and discussed.

The significant intercept indicates that when all predictors are set to 0, the risk of relapse is greater than zero. We do not find this finding to be particularly pertinent to the current study.

The sentences in lines 213-217 could be moderated since they are too speculative, and unsupported by the literature in the manuscript.

We felt it appropriate to provide some feasible explanation for the increase in first-time admissions during the pandemic. We hope we have made it clear that this explanation is speculative by the use of the following wording: “It is unclear what drove this trend…it is possible…may have driven…”

Round 2

Reviewer 1 Report

Thank you for the revisions.